# Recent Advances in Locoregional Therapy of Hepatocellular Carcinoma

**DOI:** 10.3390/cancers15133347

**Published:** 2023-06-26

**Authors:** Anna Podlasek, Maheeba Abdulla, Dieter Broering, Khalid Bzeizi

**Affiliations:** 1Tayside Innovation MedTech Ecosystem (TIME), University of Dundee, Dundee DD1 4HN, UK; podlasek.a@gmail.com; 2Precision Imaging Beacon, Radiological Sciences, University of Nottingham, Nottingham NG7 2RD, UK; 3Salmaniya Medical Complex, Arabian Gulf University, Manama 323, Bahrain; 4Department of Liver Transplantation, King Faisal Specialist Hospital and Research Center, Riyadh 11211, Saudi Arabia

**Keywords:** hepatocellular carcinoma, liver neoplasms, radiofrequency ablation

## Abstract

**Simple Summary:**

Hepatocellular carcinoma (HCC) is the fifth most common cancer worldwide and the second most common oncological reason for death. Liver resection and transplantation are considered the only potential cure options for HCC. The majority of patients, however, are late in presentation and, therefore, are considered non-suitable for surgery at the time of diagnosis. Locoregional therapies are becoming integral to its management along with systemic therapies. This review discusses the role and the advances of locoregional therapies in HCC management.

**Abstract:**

Hepatocellular carcinoma (HCC) is responsible for 90% of primary hepatic cancer cases, and its incidence with associated morbidity and mortality is growing worldwide. In recent decades, there has been a revolution in HCC treatment. There are three main types of locoregional therapy: radiofrequency ablation, transarterial chemoembolisation, and transarterial radioembolisation. This article summarises recent advances in locoregional methods.

## 1. Introduction

Hepatocellular carcinoma (HCC) is responsible for 90% of primary hepatic cancer cases. Its incidence is growing; currently, it is the fifth most common cancer worldwide, accounting for over 800,000 new cases in 2018 [1]. According to Cancer Today by WHO, it is the second most common oncological reason for death, with 50% of cases occurring in East Asia [1]. Its incidence tripled between 1980 and 2020 in the United States [2]. In Asia and Africa, HCC is usually associated with hepatitis B, whereas, in Europe, Japan, and the US, it is more often associated with hepatitis C, non-alcoholic fatty liver disease (NAFLD) and chronic alcohol abuse [1,3,4]. Other factors associated with HCC are genetic haemochromatosis, tyrosinosis, alpha-one antitrypsin deficiency, and primary biliary cirrhosis [3,5].

The diagnosis of HCC Is based on a combination of clinical, laboratory, radiographic, and histopathologic features [5]. The imaging diagnosis is based on the detection of the lesion’s vascularity [6]. Advanced imaging techniques, such as contrast-enhanced ultrasound (CEUS) and magnetic resonance imaging (MRI), have shown promising results in detecting and characterising HCC [6]. CEUS utilises microbubble-based contrast agents to provide real-time imaging of the tumour vasculature, allowing for improved lesion detection and differentiation from non-malignant liver lesions. On the other hand, MRI offers multiparametric imaging capabilities, including dynamic contrast-enhanced and diffusion-weighted imaging, enabling better tumour characterisation and assessment of treatment response [6,7] (Figure 1). Moreover, molecular imaging techniques, such as positron emission tomography (PET) using tracers like fluorodeoxyglucose (FDG), have shown potential in assessing HCC metabolic activity and predicting prognosis [6]. Recent advancements in imaging technology have also facilitated the integration of artificial intelligence (AI) algorithms to aid in diagnosing and staging HCC. These AI-based approaches leverage machine learning techniques and large datasets to improve the accuracy and efficiency of HCC diagnosis, allowing for earlier detection and intervention [8].

The tumour typically starts as a small nodule and grows during the asymptomatic phase [3]. It doubles in a median of 6 months [3]. The 1 year survival is 50–90% among untreated patients with Child–Pugh A and only 20% with Child–Pugh C [3]. The 5 year survival is low, at less than 20% worldwide [1]. Treatment is challenging as it depends on the tumour burden and the level of associated liver cirrhosis [1]. Unfortunately, despite the availability of targeted screening for HCC among high-risk groups and improvements in the prevention and treatment of risk factors, such as hepatitis B/C or NAFLD, mortality rates continue to rise [3,5,10,11]. Only 10–30% of HCC patients are candidates for surgical treatment—a curative option—because most cancers are recognised at an intermediate or advanced stage [12,13,14]. However, adding biochemical markers—such as alpha fetoprotein—significantly increases the early detection of HCC in clinical practice [15].

Globally, multiple staging systems are used to select the best treatment option for patients. The first one is the Okuda staging system, which is based on three factors: liver functional status (albumin, ascites, and bilirubin) and tumor stage (more or less than 50% of liver area involved). It is used in Japan and other countries. The second is the Barcelona Clinic Liver Cancer (BCLC) system, comprising tumour stage, liver function, and physical status. This system has been widely adopted in Europe for HCC staging and treatment [5]. Thirdly, the mUICC staging system, adopted by Korea, is based on the number of tumours, the diameter of the largest tumour, and vascular or bile duct invasion [5].

The stages of HCC vary worldwide in their presentation. In the UK, patients usually present with advanced disease, which is most often detected among people with already abnormal liver function. In contrast, 80% of HCC cases in Japan are detected when asymptomatic due to widespread screening of all people with liver cirrhosis [3]. At the time of diagnosis, 75% of HCC nodules are inoperable [16,17]. When tumours have not expanded outside the liver, locoregional treatments are applied to downstage and increase the number of liver transplant candidates or improve outcomes of patients undergoing liver resection [18,19,20,21]. The potential increase in early-stage detection based on imaging and biochemical markers may lead to increased utilisation of locoregional therapies, which currently play a leading role in 50–60% of HCC treatments [4]. The choice of liver transplantation, resection, percutaneous ablation, transarterial chemoembolisation (TACE), and/or radioembolisation treatment largely depends on tumour burden and location, as well as comorbidities [5,22]. Systemic therapy is used in moderate and advanced diseases. Classical oncological treatments, such as cytotoxic chemotherapy and hormonal therapy, have not proven successful in hepatic cancer [2]. In recent years, multiple immunotherapy options and drugs have become available [2]. The first systemic treatment for HCC was sorafenib—a multi-kinase inhibitor [2,4,22]. Around 50–60% of HCCs are managed primarily by locoregional therapies, defined as imaging-guided liver tumour-directed procedures [4]. They can be based on local ablation or intraarterial technique. The primary aim is to prolong survival by decreasing or, if feasible, eliminating the burden of hepatic tumours [4]. Patients with advanced diseases and those in the terminal stage should receive the best supportive, palliative care [5].

The treatment algorithm for HCC is constantly changing, mainly driven by the expansion of criteria for hepatic resection, advancement of locoregional and radiation therapies, and novel systemic therapies [5].

Optimal management of liver cancers depends on a multidisciplinary approach, with input and collaboration from diagnostic radiology, pathology, hepatology, transplant surgery, surgical oncology, medical oncology, radiation oncology, and interventional radiology to achieve individualised and evidence-driven patient care. Patient preferences should also be taken into consideration [7,10].

Current guidelines recommend 6 monthly surveillance of high-risk patients with ultrasound [23,24]. Further research is ongoing to optimise follow-up pathways, especially regarding MRI-based imaging [6,7]. mRECIST has become a standard tool for measuring radiological endpoints that are added to the standard cancer overall survival rates [25].

This review aims to present the up-to-date status of locoregional therapies for HCC.

## 2. Radiofrequency Ablation (RFA)

Radiofrequency ablation was introduced in the 1990s as a treatment for osteoid osteomas [26]. It is now considered the standard treatment option among local ablative techniques for very-early-stage hepatic tumours (<2 cm) and for early-stage tumours that were disqualified from the surgical approach [4]. RFA has often been deemed a curative treatment modality, with a 5 year overall survival rate of around 40–70% [2,12,27]. It is also considered the most promising locoregional treatment [28,29,30,31,32]. The electrodes are inserted into pathological tissue, and, by delivering high-frequency alternating currents, they induce coagulative necrosis and tissue desiccation [28,29,30,31,32]. The major advantages of RFA are the potential for repeatability and safety for people with significant medical comorbidities due to the lack of a need for general anaesthesia [28,33]. There is also moderate evidence for using microwaves for ablation, and low evidence for using cryoablation and irreversible electroporation [4]. 

Local tumour progression post RFA is the Achilles heel of this well-established treatment modality [12,34]. The 5 year tumour recurrence has been reported to be as high as 80%. RFA also suffers from the following limitations: ablation volume up to 5 cm, limitations related to tumour localisation (i.e., hilar or subphrenic), heat-sink effect, spreading by intratumoral pressure during RFA, and tumour seeding [28,35,36,37,38].

Every medical procedure has inherent complication risks. RFA can be complicated by severe haemorrhage, RFA needle-track seeding, abscess formation, perforation of the gastrointestinal tract, liver failure, biloma, biliary stricture, portal vein thrombosis, and haemothorax or pneumothorax requiring drainage. It has been reported that complications affect 0.6–8.9% of procedures [28,39,40]. It is worth noting that the departments treating larger numbers of patients per month had a smaller number of complications and deaths [28,41].

There are conflicting reports in the literature comparing RFA to local surgical resection. Nevertheless, local surgical resection provides better long-term oncological outcomes [42,43].

Usually, RFA is performed under ultrasonographic guidance. Recently, six reported studies compared RFA using intraprocedural CT/MRI fusion imaging versus the standard of treatment. They suggested using fusion imaging to treat large tumours in difficult anatomical positions [44].

Advanced imaging with CT or MRI is typically used to assess treatment efficacy [28]. It is separated into the following categories:Grade A—absolutely curative with 5 mm ablative margin around the entire tumour.Grade B—relatively curative, mostly as grade A with some places with the lower margin.Grade C—an incomplete ablative margin around the tumour, although no residual tumour is apparent.Grade D—absolutely noncurative; the tumour was not completely ablated [28,45].

It was reported that liver ultrasound elastography with liver stiffness could be a reliable tool for predicting recurrence after RFA [46].

RFA is often compared with microwave frequency ablation, as they are primary types of percutaneous thermal ablation. Recent summaries of studies comparing those two techniques found little to no difference in their efficacy and safety [47,48,49,50,51,52,53].

Unanswered questions remain about combination techniques. A meta-analysis of 854 patients suggested that adding percutaneous ethanol injections improves overall survival; however, the evidence is heterogeneous [54]. A network meta-analysis of 3675 patients with advanced HCC revealed that the RAF with hepatic arterial infusion chemotherapy (HAIC) achieved the highest probability of 1 year overall survival and overall response rate [55]. TACE combined with RFA or MWA can provide significantly better overall survival (HR, 0.50, 95% confidence interval [CI]: 0.40–0.62), progression-free survival (HR, 0.47, 95% CI: 0.37–0.61), and local tumour control (OR, 0.36, 95% CI: 0.24–0.53) than TACE monotherapy for patients with intermediate-stage HCC, without increasing the risk of major complications (OR, 1.26, 95% CI: 0.74–2.16) [56]. Moreover, TACE + RFA offer comparable oncologic outcomes in patients with HCC compared to surgical resection and with the added benefit of lower morbidity [57].

There is continued effort to identify the best treatment technique for HCC. A study by Kwak et al. compared percutaneous and laparoscopic RAF for HCC in the subphrenic region. The laparoscopic approach resulted in fewer local tumour progressions and increased overall survival; therefore, it is proposed as a method of choice [58].

Within the last 3 years, we identified nine randomised controlled trials involving RFA for HCC. They are summarised in Table 1.

## 3. TACE

TACE involves the injection of chemotherapy into liver tumours with a microembolus effect using iodised oil-based emulsion (lipiodol oil) to achieve arterial branch closure supplying the tumour in addition to medicinal suppression of tumour growth [21,67,68]. In 1972, the first surgical ligation of the hepatic artery with the consecutive injection of 5-fluorouracil to the portal vein was used to treat a liver tumour, which showed that the approach of blood interruption and local chemotherapy was safe. The development of an endovascular approach promptly followed it [69]. Today, an interventional radiologist enters the vascular system via the femoral approach, and then inserts the instruments to branch the hepatic artery supplying the tumour by navigating through the abdominal aorta, celiac trunk, and common hepatic artery.

TACE is the standard of care for intermediate-stage lesions (a multinodular liver-only disease in asymptomatic patients with compensated liver function). It usually contributes to the 2–2.5 year survival rate [3,4]. TACE can produce tumour necrosis and affects survival in selected patients with good liver reserve [3]. With preserved liver function, the risk of liver failure after c-TACE for HCC with portal vein invasion is acceptably low [1]. There is no consensus on optimal chemotherapeutic agents and no standardisation worldwide [1,70]. When used with lipiodol, there is an improvement in symptoms of pain and bleeding from HCC [3]. Neoadjuvant TACE can be used for patients with longer expected waiting list times for liver resection (specifically >6 months) or postoperatively in patients with a high risk of HCC recurrence [71,72,73,74].

A higher incidence of systemic adverse effects is connected with TACE due to the use of oil-based substances [68,75]. To mitigate this problem, TACE with drug-eluting beads (DEB-TACE) has been developed. It provides more selective and controlled drug delivery with microspheres [68,76,77,78]. Comparing those two treatment modalities for unresectable or recurrent HCC directly, there is no strong evidence of its increased efficacy, but it is associated with fewer side-effects [68].

There is ongoing research into clinical prognostication and patient selection for TACE. High pre-treatment albumin/bilirubin grade and aspartate aminotransferase-to-platelet index are associated with poorer outcomes [79,80]. Age, diabetes mellitus (DM), and the number of TACE sessions are risk factors for acute kidney injury—which increases mortality 4.74-fold—in patients with HCC after TACE [81]. Recently, an albumin-based algorithm was proposed [82].

There is a risk of incomplete treatment response after TACE, especially in large tumours, which are difficult to access. External beam radiotherapy provides favourable local control, but further systemic treatment could be required to improve overall survival [83]. Combining TACE with microwave ablation MWA improves 1, 2, and 3 year overall survival when compared to TACE alone for liver tumours greater than 5 cm [84,85].

There are no established imaging markers used for the prediction of TACE response. However, the delta of ADC values on MRI imaging higher than 20% facilitates early objective response to treatment [86,87]. To assess the presence of residual tumours, contrast-enhanced ultrasound can also be used. Its sensitivity is 0.85, specificity is 0.94, and accuracy is 93.5% [88]. There are many developments in post-procedure prognostication, and the optimal cut-off points in predicting the complete response of target lesions were a 52% ALT increase and a 46% AST increase after cTACE compared to the pre-treatment values [89].

The best intraarterial approach for unresectable HCC remains elusive. A network meta-analysis of 55 RCTs compared results of 5763 diverse patients among bland transarterial embolisation (TAE), cTACE, DEB-TACE, or transarterial radioembolisation (TARE), either alone or combined with adjuvant chemotherapy, local liver ablation, or external radiotherapy. All embolisation strategies improved survival, with TACE + external radiation/liver ablation achieving the highest [90]. Another study suggested the superiority of DEB-TACE over other treatment strategies [91].

Within the last 3 years, we identified 19 randomised controlled trials involving TACE for HCC. They are summarised in Table 2.

## 4. Transarterial Radioembolisation (TARE), Also Known as Selective Internal Radiation Therapy (SIRT)

Liver tissue is very sensitive to radiation. The main problem with external beam radiotherapy was that it had to pass through the healthy tissue, causing its destruction. Intraarterial therapy became a solution to this problem [10]. TARE involves an injection of β-emitting yttrium-90 (Y90), holmium-166 (166Ho) integrated inside the glass matrix or on the surface of the resin microspheres, or metuximab-131 [21,111,112,113,114,115,116]. TARE can be performed with whole-liver treatment, as well as lobar or segmental approaches (the more distal catheter placement, the more localised the technique) [117].

TARE works by inducing necrosis and delaying tumour progression [118,119,120,121,122,123]. It is widely known that patients with HCC and portal vein thrombosis (PVT) are not amenable to TACE due to the high risk of ischemia and liver failure [5,24,124]. In particular, in this subset of patients, TARE provided competitive, if not more favourable, results compared to sorafenib [124,125,126]. Only limited HCC patients are responsive to immune checkpoint inhibitors, and a combination of these with RT may enhance the immune response; this phenomenon is named the systemic therapy augmented by radiotherapy (STAR) effect [12,127].

TARE appears to be a safe alternative treatment to TACE with a comparable complication profile and survival rates [21]. However, despite these undoubted advantages, a non-negligible proportion of advanced HCC patients still do not benefit from TARE, thus calling for more effective therapeutic regimens [124]. As combining systemic agents with locoregional treatments might represent a therapeutic tool in the armamentarium of hepato-oncology, there is no evidence that the addition of sorafenib prolongs survival or delay disease progression among HCC patients undergoing TARE [124].

TARE is well known to potentially lead to serious adverse events and suffers from a narrow safety profile, which limits its worldwide use despite favourable efficacy outcomes and cost-effective benefits [120,124]. It can lead to postradioembolisation syndrome (fatigue, nausea, vomiting, abdominal pain, and cachexia), radioembolisation-induced liver disease (jaundice, ascites, hyperbilirubinemia, and hypoalbuminemia 2–4 weeks post treatment), portal hypertension, and biliary complications (biliary strictures or cholecystitis), as well as radiation pneumonitis, gastrointestinal ulcers, and vascular injury [128,129,130]. However, in a meta-analysis of 1652 patients based on 11 studies, Y90-TARE not only improved 2 year overall survival and objective response among observational studies [130], but was also associated with fewer adverse events compared to TACE [90,130,131].

The current evidence suggests that there is a dose–response relationship for HCC tumours, with the best current evidence for the target mean dose of 100–250 Gy [132]. There is a need for the development of reporting standards and dose-dependent guidelines [132].

The reported economics of TARE as an interventional modality of HCC is largely variable. Overall, it appears cost-effective as a short- and long-term treatment of intermediate-advanced HCC [133].

Within the last 3 years, we identified four randomised controlled trials involving TARE for HCC. They are summarised in Table 3.

## 5. Conclusions

Locoregional therapies have established their place in the HCC management algorithm. RFA has the potential for repeatability and safety for patients with significant medical comorbidities. The primary concerns with this procedure remain local tumour progression post RFA, needle-track seeding, and abscess formation.

TACE is the standard of care for intermediate-stage lesions. It can produce tumour necrosis and improve survival in patients with good liver reserve. Neoadjuvant TACE can be used for patients with longer expected waiting list times for liver surgery (resection or transplant). TACE with drug-eluting beads (DEB-TACE) provides more selective and controlled drug delivery with microspheres than cTACE. Although DEB-TACE is associated with fewer side-effects, it has no strong evidence of increased efficacy compared to cTACE.

TARE provides a safe alternative treatment to TACE with a comparable complication profile and survival rates. TARE is well known to potentially lead to serious adverse events and suffers from a narrow safety profile, which limits its worldwide use despite favourable efficacy outcomes and cost-effective benefits. A dose–response relationship exists for HCC tumours with the best current evidence for the target mean dose of 100–250 Gy. However, there is a need to develop reporting standards and dose-dependent guidelines. 

More research is needed to identify the optimal locoregional HCC treatment, better identify the early predictive factors, and develop an individualised treatment regimen. With the availability of the checkpoint immunotherapy modalities, the interest in combining locoregional and systemic therapies has resurfaced, and results of the ongoing trials of these combinations are eagerly awaited.

## Figures and Tables

**Figure 1 cancers-15-03347-f001:**
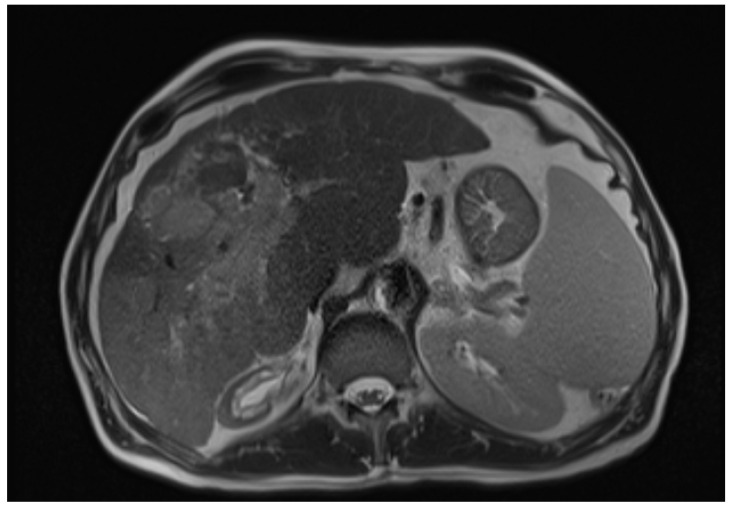
Axial T2-weighted MRI image of hepatic cirrhosis with HCC [9].

**Table 1 cancers-15-03347-t001:** Summary of the recent randomised controlled trials on radiofrequency ablation (RAF).

Author, Year, Trial Name	Population	Sample Size	Intervention	Comparison	Outcome
Hendriks, 2022, HORA EST HCC Trial [59]	HCC patients with a solitary lesion 2–5 cm, or a maximum of 3 lesions of ≤3 cm each	NA	Day 1 post RFA: selective infusion ^99m^Tc-MAA), days 5–10 post RFA: ^166^Ho-MS administration	60 Gy, 90 Gy, and 120 Gy of ^166^Ho-MS	Perfused liver volume; final outcome not yet available
Radosevic, 2022 [60]	HCC and patients with metastatic disease with 1.5–4 cm tumours, suitable for ablation	82	Ablation: MWA and RFA	Between MWA and RFA in S.L.R., T.S., LTP	After a median 2-year follow-up, MWA vs. RFA: SLR: 0.5 vs. 0.5 *p* = 0.229 TS: 98% vs. 90% *p* = 0.108 LTP: 21% vs. 12% *p* = 0.238
Suh, 2021 [61]	Eligible patients for RFA with H.C.C.	73	RFA: conventional or no NT-RFA using twin internally cooled wet electrodes in the bipolar mode	Between NT-RFA and RFA groups in LTP rates	The 1 and 3 year cumulative LTP rates were 5.6% in the NT-RFA group, and they were 11.8% and 21.3%, respectively, in the conventional RFA group (*p* = 0.073, log-rank)
Bockonry, 2022 [62]	For HCC tumours sized 3.5–7 cm	20	Priming with 400 mg sorafenib BD for 10 days prior to RFA or placebo	Priming versus placebo in volume and diameter of the RFA coagulation zone	No increase in ablation volume/diameter; decreased blood perfusion to the tumour by 27.9% (*p* = 0.01)
Kim, 2021 [63]	≤2 recurrent H.C.C. of <3 cm	144	PBT or RFA	PBT vs. RFA in LPFS	PBT showed LPFS values that were noninferior to those for RFA
Choi, 2020 [64]	Recurrent HCC after locoregional treatment	77	RFA: TICW, bipolar, using twin internally cooled wet electrodes or SC: switching monopolar RFA, using separable clustered electrodes	TICW-RFA vs. SC-RFA in minimum diameter of the ablation zone per unit ablation time	No significant Difference
Choi, 2021 [63]	HCC	80	RFA; DSM: dual-switching monopolar; SSM conventional single-switching monopolar	DSM-RFA vs. SSM-RFA in minimum diameter of the ablation zone per unit ablation time	No significant Difference
Chong, 2020, McRFA trial [65]	HCC suitable for local ablation	93	Ablation: MWA and RFA	MWA vs. RFA in treatment-related morbidity, as well as overall and disease-free survival	No significant difference in the treatment-related morbidity or overall and disease-free survival; MWA had a significantly shorter overall ablation time when compared with RFA (12 min vs. 24 min, *p* < 0.001)
Paul, 2020 [66]	HCC < 5 cm	55	PAAI, RFA	PAAI vs. RFA in tumour response and survival rate	Similar efficacy

^99m^Tc-MAA, technetium-99-labeled microalbumin aggregates; ^166^Ho-MS, holmium-166; MWA, microwave ablation; SLR, short-to-long diameter ratio of ablation zone; TS, Primary technical success; LTP, cumulative local tumour progression; NT-RFA, no-touch RFA; PBT, proton beam therapy; LPFS, 2 year local progression-free survival; PAAI, percutaneous acetic acid.

**Table 2 cancers-15-03347-t002:** Summary of the recent randomised controlled trials on TAC.

Author, Year, Trial Name	Population	Sample Size	Intervention	Comparison	Outcome
Zhang, 2022 [92]	TACE for HCC	130	Femoral or radial approach (radial without the closure device)	Technical success rate, crossover rate, contrast agent dose, fluoroscopy/procedure time, air kerma, dose–area product, length of hospital stay, total cost, incidence and severity of adverse events, overall discomfort, general health, physical/social/emotional function, and mental health	TRA instead of TFA can improve patient satisfaction without compromising procedural variables and safety
Jiang, 2022 [93]	TACE for HCC	120	Femoral or radial approach with foot or head-first position	Radiation dose at 7 anatomical sites of the operator	TACE via the left TRA, with patients in the feet-first position, reduced the radiation dose received by the operator
Zhu, 2022 [94]	TACE for unresectable HCC	72	TACE with a distinct chemotherapeutic regimen	A chemotherapeutic regimen of dicycloplatin alone (group A1), dicycloplatin plus epirubicin (group A2), or epirubicin alone (group B)	TACE with dicycloplatin alone or plus epirubicin was comparably safe and well tolerable as epirubicin alone; significant improvements in ORR and DCR when dicycloplatin was applied, and prolonged PFS when dicycloplatin plus epirubicin was applied compared with epirubicin alone
Dhondt, 2022 [95]	Intermediate-stage unresectable HCC with ECOG 1 or early-stage HCC not eligible for surgery or thermoablation	72	90Y glass TARE was compared with doxorubicin DEB-TACE	Time to overall tumour progression	90Y glass TARE superior tumour control and survival compared with doxorubicin DEB-TACE
Llovet, 2022, LEAP-012 [96]	HCC localised to the liver without portal vein thrombosis and not amenable to curative treatment, ≥1 measurable tumour per Response Evaluation Criteria in Solid Tumours 1.1 (RECIST 1.1), ECOG 0 or 1, Child–Pugh class A, and no previous systemic treatment for HCC	950	Lenvatinib once daily plus pembrolizumab every 6 weeks plus TACE or placebos plus TACE	PFS, objective response rate, disease control rate, response duration and progression time, and safety	Study ongoing
Chen, 2022, GALNT14 [97]	Intermediate-stage HCC patients	84	GALNT14-rs9679162 genotyping before TACE and division into (1) “TT” genotype receiving TACE, (2) “non-TT” genotype (“GG” or “GT”) randomised into TACE or TACE + sorafenib groups	Time to complete response, time to TACE progression, PFS, and OS	Sorafenib + TACE for “non-TT” partially overcame the genetic disadvantage on treatment outcomes in terms of time to complete response, time to TACE progression, and progression-free survival
Zhang, 2022 [98]	Massive HCC	92	TACE or TACE + camrelizumab	Clinical efficacy, adverse events, liver function, and AFP, CEA, and CA19-9 levels before and after treatment	Camrelizumab + TACE can significantly improve liver function and enhance the treatment effect
Zhang, 2022 [99]	HCC with PVTT	627	TACE or liver resection or sorafenib	OS	Liver resection optimal for type I and II PVTT, TACE optimal for type III PVTT, and sorafenib optimal for type IV PVTT
Aramaki, 2021, ACE 500 [100]	Liver-confined HCC, ECOG 0–2, and Child–Pugh class A/B	455	TACE with cisplatin vs. TACE with epirubicin	OS	Cisplatin is not significantly superior to epirubicin in TACE for patients with HCC
Fu, 2021 [101]	HCC patients resistant to TACE with doxorubicin	170	TACE with doxorubicin vs. TACE with bleomycin	Objective response rate and post-procedure complications	Bleomycin can be a safe and effective second-line chemotherapeutic agent for HCC patients unresponsive to TACE with doxorubicin
Ding, 2021 [102]	HCC with PVTT	64	TACE with lenvatinib vs. TACE with sorafenib	Time to progression, objective response rate, and toxicity	TACE plus lenvatinib was safe and well tolerated, and had favourable efficacy versus TACE plus sorafenib in patients with advanced HCC with PVTT and large tumour burden
Yang, 2021 [103]	HCC	291	TACE with or without FZJDXJ	1 year OS and PFS	FZJDXJ combined with TACE therapy significantly prolonged OS and PFS and reduced the mortality rate of HCC patient
Pan, 2021 [104]	HCC	50	Post-TACE: placebo vs. Chaihu Guizhi decoction	Incidence of PES	study ongoing
Bessar, 2021 [105]	HCC	28	TACE with doxorubicin: 50 mg vs. 100 mg	Incidence of PES, free time to PES, changes in laboratory results, tumour response at 1, 3, and 6 months after TACE, and OS	50 mg doxorubicin was associated with fewer PES without effects on tumour response or OS
Zaitoun, 2021 [106]	HCC >3–<5 cm	265	TACE vs. MWA vs. TACE+MWA	Treatment response, adverse events, and AFP	TACE + MWA is safe, well-tolerated, and more effective than TACE or MWA alone for the treatment of HCC >3–<5 cm
Gjoreski, 2021 [107]	Unresectable HCC	60	TACE vs. DEM-TACE	12 and 24 month OS,	No significant difference in overall OS, or adverse events; TACE was associated with more severe PES and DEM-TACE with a shorter in-hospital stay
Guo, 2020 [108]	Advanced HCC	117	TACE + HAIC + oral S1 vs. TACE + HAIC	PFS, OS, objective response rate, disease control rate, and safety	No improvements in tumour response rates, PFS, OS, or adverse events were observed with the addition of S-1 to TACE/HAIC in advanced HCC
Turpin, 2020, PRODIGE 16 [109]	Unresectable HCC	78	Doxorubicin-TACE + sunitinib vs. doxorubicin-TACE + placebo	Bleeding or liver failure	TACE plus sunitinib in the first-line therapy for patients with HCC not suitable for surgical resection was feasible
Kudo, 2020, TACTICS [110]	unresectable HCC	156	TACE + placebo vs. TACE + sofafenib	PFS, OS, time to untreatable progression, transient deterioration to Child–Pugh C, and appearance of vascular invasion/extrahepatic spread	TACE plus sorafenib significantly improved PFS

HCC, hepatocellular carcinoma; TRA, transradial approach; TFA, transfemoral approach; ORR, objective response rate; DCR, disease control rate; PFS, progression-free survival; ECOG, Eastern Cooperative Oncology Group performance status; 90Y glass TARE, yttrium 90 transarterial radioembolisation; OS, overall survival; DEB-TACE, drug-eluting bead transarterial chemoembolisation; AFP, alpha-fetoprotein; CEA, carcinoembryonic antigen; CA19-9, carbohydrate antigen 19-9; PVTT, portal vein thrombus; FXJDXJ, The Fuzheng Jiedu Xiaoji formulation; PFS, progression-free survival; PES, postembolisation syndrome; MWA, microwave ablation; DEM, drug-eluting microspheres; HAIC, hepatic arterial infusion chemotherapy.

**Table 3 cancers-15-03347-t003:** Summary of the recent randomised controlled trials on TARE.

Author, Year, Trial Name	Population	Sample Size	Intervention	Comparison	Outcome
Hendriks, 2022, HORA EST HCC Trial [59] *	HCC patients with a solitary lesion 2–5 cm, or a maximum of 3 lesions of ≤3 cm each	NA	Day 1 post RFA: selective infusion ^99m^Tc-MAA), day 5–10 post RFA: ^166^Ho-MS administration	60 Gy, 90 Gy and 120 Gy of ^166^Ho-MS	Perfused liver volume; final outcome not yet available
Dhondt, 2022 [95] **	Intermediate-stage unresectable HCC with ECOG 1 or early-stage HCC not eligible for surgery or thermoablation	72	^90^Y glass TARE was compared with doxorubicin DEB TACE	Time to overall tumour progression	^90^Y glass TARE superior tumour control and survival compared with doxorubicin DEB TACE
Pereira, 2021, SARAH [134]	Locally advanced or inoperable HCC	285	TARE vs. sorafenib	HRQoL	HRQoL was preserved longer with TARE than with sorafenib
Eisenbrey, 2021 [135]	HCC	28	TARE vs. TARE with ultrasound-triggered microbubble destruction	Safety and preliminary efficacy	Microbubbles have an excellent safety profile in this patient population and appear to result in improved hepatocellular carcinoma treatment response

HCC, hepatocellular carcinoma; HRQoL, health-related quality of life. * Also presented in Table 1, ** also presented in Table 2.

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
