# Peer review of "Recent Advances in Locoregional Therapy of Hepatocellular Carcinoma"

_cancers, 2023, doi:10.3390/cancers15133347_

Round 1

Reviewer 1 Report

Very interesting and well comprehensive review. My comments:

1) Some figures would improve the quality of the paper.

2) English grammar requires some polishing/minor editing

3) Reference 41 is only an editorial. I recommend to replace it with a full text paper on the same topic: PMID: 33339274 )

4) The authors should discuss the potential prognostic predictors in patients undergoing loco-regional treatments, specifically TACE (in this regard cite the recent series PMID: 34683182)

5) References 106 and 107 are unclearly reported. 

Minor revision

Author Response

Many thanks to all reviewers for taking the time to provide useful and practical suggestions. We have implemented the changes, thus improving the manuscript quality. Please find the attached individual replies to the points raised by Reviewer 1:

Reviewer 2 Report

TITLE

In my opinion the title is interesting and attractive.

KEYWORDS

Authors did not correctly report all keywords from MeSH Browser. In particular, for example, I checked “transarterial chemoembolization,” on MeSH Browser and I did not find this KW. This is important, in my personal opinion, in order to increase the traceability of this paper (and consequently the possibility of the Journal to be cited by Readers and Stakeholders). I suggest to check all KWs and change those not reported on MeSH Browser.

ABSTRACT

The abstract is short but well structured and properly reflects the main text highlighting only the most important aspects of this paper, consequently no major adjustments are needed.

INTRODUCTION

In my opinion, the introduction could be improved by the Authors.

The authors could describe with greater attention some aspects of the main topics.

In my opinion the sentences between lines 39 and 44 (“The alpha-fetoprotein….. of 6 months.[3]) are not focused on the topic of this paper.

Conversely, the authors did not report the state of the art of the imaging diagnosis of HCC. For example, they can report the role of US, CEUS, CT and MRI, citing [Contrast Agents for Hepatocellular Carcinoma Imaging: Value and Progression. Front Oncol. 2022;12:921667. Published 2022 Jun 2. doi:10.3389/fonc.2022.921667] and [Proposal of a new diagnostic algorithm for hepatocellular carcinoma based on the Japanese guidelines but adapted to the Western world for patients under surveillance for chronic liver disease. J Gastroenterol Hepatol. 2016;31(1):69-80. doi:10.1111/jgh.13150].

In particular, the authors should analyze an important theme that correlates a good and early diagnosis of HCC with the possibility of being able to apply locoregional treatments. The authors Could discuss this future scenario: in Europe, some countries in Asia and USA, the surveillance program and also the future strategy to improve the surveillance program [Non-enhanced magnetic resonance imaging as a surveillance tool for hepatocellular carcinoma: Comparison with ultrasound. J Hepatol. 2020;72(4):718-724. doi:10.1016/j.jhep.2019.12.001   -----    Proposal of a new diagnostic algorithm for hepatocellular carcinoma based on the Japanese guidelines but adapted to the Western world for patients under surveillance for chronic liver disease. J Gastroenterol Hepatol. 2016;31(1):69-80. doi:10.1111/jgh.13150], will allow to overcome the ultrasound limitations in the detections of HCC at an early and very early stages.

In fact, actually ultrasound identify 4 out 10 patients in very early or early stages [Surveillance Imaging and Alpha Fetoprotein for Early Detection of Hepatocellular Carcinoma in Patients With Cirrhosis: A Meta-analysis. Gastroenterology. 2018;154(6):1706-1718.e1. doi:10.1053/j.gastro.2018.01.064]. The first consequence will be the detection of even-increasing number of lesions in very early and early stage (small lesions). Could the Authors discuss these themes, cite the papers and report their ideas about these possible scenarios? For example: could these new scenario improve the utilization of locoregional treatments of HCC?

Furthermore, please, could the Authors introduce the aim of their review at the end of the introduction section?

Radiofrequency ablation (RFA)

Authors did not describe different points: cryoablation and irreversible electroporation [Nat Rev Gastroenterol Hepatol. 2021 May;18(5):293-313. doi: 10.1038/s41575-020-00395-0]?

It is important, in my opinion, to cite mRECIST.

TACE

It is well known that one of the limitations of TACE is its standardization. Recently, a paper tried to standardize TACE [Ann Hepatol. 2021 May-Jun;22:100278. doi: 10.1016/j.aohep.2020.10.006. Epub 2020 Oct 29. PMID: 33129978.]. please, could you discuss this theme?

A multitude of TACE-specific staging systems have been developed to use in pre-procedural phase, such as albumin-based liver reserve models [Cancers (Basel). 2023;15(7):1925. Published 2023 Mar 23. doi:10.3390/cancers15071925]. Furthermore, also postprocedural parameters have been investigated as outcome predictors of TACE’s efficacy, such as the Transient Hypertransaminasemia after TACE [J Pers Med. 2021;11(10):1041. Published 2021 Oct 17. doi:10.3390/jpm11101041].

did the authors use these pre- and post-procedural evaluations? Please, may the authors mention it?

TARE

It is clear.

In my opinion the authors could introduce some possible novelties such as [J Nucl Med 2022;63:556–559].

CONCLUSIONS

The discussion section is ok.

REFERENCES

References reflect the style showed in the “Instruction for Authors”.

Author Response

Many thanks to all reviewers for taking their time to provide useful and practical suggestions. We have implemented the changes, thus improving the manuscript quality. Please find the attached individual replies to the points raised by Reviewer 2:
